# Impact of Reduced Image Noise on Deauville Scores in Patients with Lymphoma Scanned on a Long-Axial Field-of-View PET/CT-Scanner

**DOI:** 10.3390/diagnostics13050947

**Published:** 2023-03-02

**Authors:** Kirsten Korsholm, Nanna Overbeck, André H. Dias, Annika Loft, Flemming Littrup Andersen, Barbara Malene Fischer

**Affiliations:** 1Department of Clinical Physiology and Nuclear Medicine, Rigshospitalet, 2100 Copenhagen, Denmark; 2Department of Nuclear Medicine and PET Centre, Aarhus University Hospital, 8200 Aarhus, Denmark; 3Department of Clinical Medicine, University of Copenhagen, 2200 Copenhagen, Denmark

**Keywords:** LAFOV, Deauville score, lymphoma, PET/CT, reconstruction methods

## Abstract

Background: Total body and long-axial field-of-view (LAFOV) PET/CT represent visionary innovations in imaging enabling either improved image quality, reduction in injected activity–dose or decreased acquisition time. An improved image quality may affect visual scoring systems, including the Deauville score (DS), which is used for clinical assessment of patients with lymphoma. The DS compares SUVmax in residual lymphomas with liver parenchyma, and here we investigate the impact of reduced image noise on the DS in patients with lymphomas scanned on a LAFOV PET/CT. Methods: Sixty-eight patients with lymphoma underwent a whole-body scan on a Biograph Vision Quadra PET/CT-scanner, and images were evaluated visually with regard to DS for three different timeframes of 90, 300, and 600 s. SUVmax and SUVmean were calculated from liver and mediastinal blood pool, in addition to SUVmax from residual lymphomas and measures of noise. Results: SUVmax in liver and in mediastinal blood pool decreased significantly with increasing acquisition time, whereas SUVmean remained stable. In residual tumor, SUVmax was stable during different acquisition times. As a result, the DS was subject to change in three patients. Conclusions: Attention should be drawn towards the eventual impact of improvements in image quality on visual scoring systems such as the DS.

## 1. Introduction

Total-body positron emission tomography (PET) and long axial field-of-view (LAFOV) PET represent visionary innovations in clinical nuclear medicine with improved sensitivity compared to standard-axial field-of-view (SAFOV) PET. These new techniques enable either improved image quality, reduction in injected activity-dose, or decreased acquisition-time [1]. The improved image quality is mainly a result of the extended axial FOV capturing more photon pairs and thus providing a higher detection efficiency and sensitivity gain of 5–10 x compared to the same detector SAFOV system [2], but also the state-of-art time-of-flight (TOF) resolution of 225 ps contributes to an increase in effective sensitivity [3].

With PET/CT structural anatomy is combined with metabolic information, and this modality is commonly used in oncology, cardiology, rheumatology, and infectious diseases. The most widely used tracer in oncology is Fluorine-18-fluorodeoxyglucose ([^18^F]FDG), a glucose analogue providing a unique means of non-invasive assessment of tumor glucose metabolism.

Malignant lymphomas comprise a heterogeneous group of cancers, and the risk of being diagnosed with lymphoma increases markedly with age. Hodgkin and Burkitt lymphomas dominate in younger ages, whereas follicular, marginal zone, mantle cell, and diffuse large B-cell lymphomas are more common with older age [4].

[^18^F]FDG PET/CT has become the standard procedure for staging of disease in patients with FDG-avid lymphomas, as [^18^F]FDG-PET/CT is superior to CT alone in delineating extent of nodal/extranodal disease, including liver, spleen, and bone marrow involvement.

During treatment of lymphoma, an interim [^18^F]FDG-PET/CT after two (1–4) cycles of chemotherapy can help assess early treatment response and thereby differentiate between patients needing escalated treatment regimens or reduced intensity protocols. In addition, [^18^F]FDG-PET/CT is used to evaluate status after end of treatment (EOT), with complete metabolic remission (CMR) at both interim and EOT predicting a better overall survival (OS) and longer progression-free-survival (PFS) [5].

In 2009, the Deauville score (DS) was introduced [6]. The DS ranges from 1 to 5 and scores the highest metabolic activity in (eventual) residual disease compared to metabolic activity in liver and mediastinal blood pool. The DS is now standard for reporting of clinical [^18^F]FDG-PET/CT scans.

With the publication of the Lugano classification in 2014 [7], non-progressive disease could be divided into CMR in case of DS of 1, 2, or 3 with FDG-uptake equal to or less than liver-uptake or partial metabolic response (PMR) with a DS of 4 or 5—with reduced uptake compared with baseline. Stable disease or no metabolic response refers to a DS of 4 or 5 with no obvious change in FDG uptake, and progressive disease to a score of 4 or 5 in any lesion with an increase in intensity of FDG uptake from baseline (and/or new FDG-avid foci consistent with lymphoma).

In recent years, new iterative reconstruction algorithms including point-spread-function (PSF) and TOF have been introduced. These new algorithms can significantly change maximal standardized uptake values (SUVmax) especially in small lesions compared to conventional reconstruction algorithms, however, only moderately affecting SUVmax in liver and vascular background [8,9,10]. Accordingly, Quak and coworkers reported that the use of PSF could increase the lesion-to-liver ratio (based on SUVmax) with up to 31% [11]. As the DS compares SUVmax in residual lymphoma to SUVmax of the mediastinal blood pool and the liver, these new reconstruction methods can have a considerable impact on the DS, as reported by various groups [12,13,14].

With the new LAFOV PET/CT-scanner systems, the increased sensitivity can be exploited to either reduce acquisition time, injected radioactivity dose, or a combination of both. Alberts et al. [1] demonstrated that their LAFOV PET/CT system could deliver images in less than 2 min with an image quality comparable to those from a SAFOV PET/CT obtained in 16 min. In addition, even shorter acquisition times (down to 0.5 min) allowed for adequate image quality with respect to lesion detection. Van Sluis et al. [15] confirmed this ability to reduce scan time with a LAFOV PET/CT and they, too, reported a markedly reduced noise in the liver with increasing scan duration—especially when the reconstruction method included PSF.

With our new LAFOV scanner, we also noticed a remarkably reduced image noise with increasing acquisition time, especially in the liver. Therefore, with this study, we seek to compare DS for different image acquisition times (90 s, 300 s, and 600 s) on the LAFOV Siemens Biograph Vision Quadra system, Siemens Healthineers, Knoxville, reconstructed with and without PSF to investigate if/how this might influence the DS in patients with lymphomas. We hypothesized that the reduced image noise, especially in the liver, could result in an increased tumor-to-liver ratio and thus a higher DS.

## 2. Materials and Methods

### 2.1. Patients

Ninety-four consecutively referred patients with lymphoma referred for [^18^F]FDG PET/CT from 1 September 2021, until 31 January 2022 were included. Patients referred for assessment of treatment response (both interim and EOT) in follicular, Hodgkin, B-cell, and T-cell lymphoma were included in the study. Patients referred for initial staging of lymphoma disease, suspicion of recurrence, or unconfirmed suspicion of lymphoma were excluded (21 patients). Approval from the local Ethics Committee was not required as the project qualifies as a quality assurance study. The study was approved by the departmental review board (Ref. no 481_21), and all patients gave written and oral consent to participate in the study.

### 2.2. PET Acquisition and Reconstruction Parameters

PET-scans were performed on a Biograph Vision Quadra PET/CT-scanner (Siemens Healthineers, Knoxville, TN, USA) in the Department of Clinical Physiology and Nuclear Medicine, Rigshospitalet, Copenhagen, Denmark according to the EANM guidelines for tumor imaging [16]. Patients fasted for 4 h before injection of 3 MBq/kg [^18^F]FDG intravenously. All PET-studies were performed approx. 60 min after tracer injection. Patients were scanned from the base of the skull to mid-thighs. PET reconstructions were performed by two different methods: Ordered Subset Expectation Maximization (OSEM) or OSEM+PSF, termed TrueX by the vendor. Both methods included TOF and were reconstructed using 4 iterations of 5 subsets into 440 by 440 matrices (1.65 mm × 1.65 mm voxel size) with a slice thickness of 2 mm matching the CT. Data were acquired in list mode with full acceptance angel, for reconstruction a maximum ring difference (MRD) of 85 was used. Gaussian post-filters of 4 mm and 2 mm were used for OSEM and OSEM+PSF, respectively, and all data were reconstructed into static time frames with duration of 90 s, 300 s, and 600 s.

Diagnostic CT with intravenously and orally administered contrast was performed when the indication was EOT; for interim treatment response, a low-dose CT was performed. Attenuation correction was based on a low-dose CT.

### 2.3. Clinical Evaluation

A team consisting of a nuclear medicine specialist and an onco-radiologist evaluated all scans for clinical purposes on a Syngo.via workstation (Siemens Healthineers) before enrollment in the study.

### 2.4. Quantitative Evaluation

For quantitation of FDG-uptake in the liver/mediastinum and in residual lymphomas, we used the medical imaging software Mirada DBx (version 1.2.0.59, Mirada Medical Limited, 2016, Oxford, UK). The metabolically most intense residual target lesion in each patient representing residual lymphoma was contoured (volume of interest, VOI); in addition, a banana-shaped VOI was drawn in the center of the right lobe of the liver and another VOI in the thoracic aorta avoiding the vessel wall and eventual calcifications (mediastinal blood pool, MBP) [5]. The VOIs were saved as DICOM Radiotherapy structure sets (RTstructs). All VOIs were outlined on OSEM+PSF 90 seconds’ reconstruction images and subsequently transferred to all other image reconstructions. The reconstructions of the two methods with three different frame durations were converted into MINC format (McConell Brain Imaging Centre, Montreal) and resampled to fit the CT slices. The RTstructs were converted into MINC format and used as a mask to retrieve the intensity values within the different VOIs. The mean, maximum, and standard deviation were retrieved for every VOI and converted into SUVs. All baseline scans were systematically reviewed to ensure initial involvement of the tumor site. No diffuse lymphoma-involvement of the liver was noticed in any of the patients.

### 2.5. Visual Evaluation

Images reconstructed with respectively OSEM (90, 300, and 600 s) and OSEM+PSF (90, 300, and 600 s) were evaluated visually on two separate days by two experienced nuclear medicine physicians. DS was assigned as DS 1 (no visible lesion and no residual uptake), DS 2 uptake ≤ mediastinal blood pool, DS 3 > mediastinal blood pool ≤ liver, DS 4 uptake > liver, DS 5 uptake markedly (2–3 times) > liver.

### 2.6. Statistics

Differences in SUVmax between different acquisition times within the same reconstruction method, and between reconstruction methods, were assessed using a one-tailed paired *t*-test. A *p*-value < 0.05 was considered statistically significant. Coefficient of variance (COV) was calculated for characterization of image noise (defined as the ratio of the standard deviation of SUV to the mean SUV in healthy liver tissue). All statistical procedures were performed using IBM^®^ SPSS^®^ Statistics.

## 3. Results

Seventy-three patients were included; however, due to missing data, five patients were omitted from further analysis, ending up with 68 patients. Clinical characteristics of the patients including age, sex, and type of lymphoma are displayed in Table 1.

### 3.1. Visual Evaluation

As expected, image noise was visually clearly reduced with increasing acquisition time, especially in liver parenchyma (Figure 1A).

DS of all patients for the different time series and reconstruction methods are displayed in Table 2.

With OSEM+PSF reconstruction, the DS differed in three patients: Patient number 21 with refractory Hodgkin’s lymphoma scored DS 4 on 90 s images and DS 5 on 300 and 600 s images due to visually clearly reduced noise in liver. Patients number 41 and 52, both with DLBCL, scored DS 3 on 90 s images and DS 4 on the longer reconstructions (Figure 1B), also due to significant visual reduction of noise in the liver (see also Appendix A).

Only one patient (no. 21) differed in DS within the different OSEM reconstructions with DS 4 at 90 s images and DS 5 at 300 and 600 s images. All other patients scored equally with OSEM reconstruction.

### 3.2. Clinical Implications

In two patients scored with DS on OSEM+PSF, the difference in DS could have an implication on further treatment, as DS 3 is considered CMR (responder), whereas DS 4 is considered PMR (non-responder). With OSEM reconstruction, no impact on further treatment was observed as only one patient differed in DS (DS 4 to DS 5, both non-responder) for all acquisition times.

### 3.3. Quantitative Evaluation

SUVmax and SUVmean of liver parenchyma for the different acquisition times for OSEM+PSF and OSEM reconstruction are displayed in Figure 2. There were significant differences in SUVmax between all acquisition times within both OSEM+PSF and OSEM reconstructions (*p* < 0.05); however, no significant differences were observed between SUVmean for the different acquisition times for either reconstruction method. The same pattern applies to MBP (see Appendix A). The tumor SUVmax in OSEM+PSF and OSEM did not differ significantly between the different acquisition times (Appendix A). Comparing tumor SUVmax between reconstruction methods, we found a significant higher SUVmax (*p* < 0.05) when using PSF. Due to less noise in the liver, the tumor SUVmax/liver SUVmax ratio increased with increasing acquisition time (Figure 3); however, the tumor SUVmax/liver SUVmean remained unchanged (Appendix A).

### 3.4. Image Noise

Coefficient of variance (COV) decreased as expected with increasing acquisition time (Figure 4). A COV < 15% is considered an acceptable image noise level for clinical interpretation [17], and for 300 s images, all scans were below this level—except for two outliers.

## 4. Discussion

In this study, we compared FDG-PET/CT scans with different acquisition times on a new generation LAFOV PET/CT scanner. The visually most convincing change in image quality was seen in the liver parenchyma with a remarkable reduction in image noise. Quantitatively, we observed that the liver SUVmax decreased significantly with longer acquisition times.

This change in SUVmax in liver parenchyma can have different implications, among others in patients with malignant lymphoma, where the DS is defined as the ratio between tumor SUVmax and liver SUVmax. We did not see any change in tumor SUVmax between different acquisition times—neither for reconstructions with PSF nor without PSF. Due to decreasing SUVmax in liver, the DS changed with longer acquisition times in three patients, when reconstruction included PSF, two of these from DS 3 to DS 4, which could have an impact on further treatment. Without PSF, DS was only subject to change in one patient, from DS 4 to DS 5.

Previously, Enilorac and coworkers [14] reported that risk stratification of patients with lymphomas was not affected by choice of reconstruction method, although DS was re-classified due to reconstruction method in 14% (I-PET) and 8.4% (EOT) of their patients. Contrasting this, Ly and coworkers [12] described that using different reconstruction methods (silicon-photomultiplier-based (SiPM) reconstruction, commercially sold as Q.Clear, versus OSEM), could have a large impact on DS as five non-responders (DS 4 and DS 5) in their study were reclassified as responders (≤D3). This was in agreement with Wyrzykowski and coworkers [13] who report that the use of Q.Clear reconstruction algorithms caused an alteration in DS in 22 cases, of which 10 cases were converted to the non-responder group, in four cases with impact on treatment strategy. This was also in agreement with our findings, where the use of PSF generated more alterations in DS than when using OSEM.

SUVmax in small lesions can differ between different reconstruction methods. This also applies to our study, where we found a significant higher tumor SUVmax in reconstructions with PSF compared to OSEM in concordance with previous studies [18,19,20].

Two studies have previously evaluated the impact of reduced scan time on DS in patients with lymphoma and both reported no change in DS, when either reducing acquisition time from 120 s to 90 s per bed position [17] or reducing total scan time from 15 min to 5 min using continuous-bed-motion [21]. However, both studies were performed on Siemens Biograph Vision systems and not on a Biograph Vision Quadra PET/CT system, which provides an axial FOV of 106 cm, a higher spatial resolution, and a remarkably increased sensitivity, probably explaining the differences. Yet, they found that reduced acquisition time led to an increase in image noise, which is in line with our results and other studies [15,22,23].

We found that liver SUVmean did not change significantly with changing acquisition time, whereas liver SUVmax decreased significantly with increasing acquisition time, explained by SUVmax being based on a single voxel and therefore more sensitive to noise. This supports previous findings and suggestions that liver SUVmean would perform better as reference instead of liver SUVmax. For example, Zwezerijnen et al. [24] report that liver SUVmean is the most robust metric against VOI size, location, reconstruction protocol, and image noise level, and they propose to use liver SUVmean as reference for tumor assessment instead of SUVmax. Others take it further and propose to replace the visual Deauville scale by a quantitative method, such as qPET or ΔSUVmax, which minimizes the confounding factors of visual assessment [25,26]. Furthermore, the aforementioned prefer using SUVmean of the liver as reference standard, as advanced images reconstruction methods may overestimate SUVmax compared to SUVmean.

Another approach using a lesion-to-liver ratio (LLR) of SUVmax in EOT-PET/CT in patients with DLCBL was investigated by Li et al. [27]. They compared the prognostic value of the DS with LLR and reported that a LLR > 1.83 exhibited higher specificity than DS 4-5 indicating superiority in defining patients with need for additional treatment after first line treatment.

Others have explored different methods for predicting event-free survival including total metabolic tumor volume (TMTV) in patients with DLBCL [28,29] and healthy organ uptake in patients with Hodgkin’s lymphoma at baseline [30], the latter exploring the inverse correlation between FDG-uptake in cerebellum and liver and TMTV, which could be explained by a metabolic theft of FDG from large tumor masses leaving less FDG available for healthy organs.

Detailed information on, e.g., tumor texture, shape, dissemination patterns or heterogeneity—also known as quantitative radiomics—can also help identifying patients at risk of relapse. In a large group of patients with DLCBL, combining the International Prognostic Index (IPI) with radiomics of metabolic tumor volume (MTV) and dissemination pattern significantly improved identification of patients with risk of relapse compared to IPI alone [31].

However, reconstruction parameters are not the only source of variation in SUV and tumor-to-liver ratios, and especially patient preparation may have an influence. It has been reported that higher blood glucose levels are associated with increased FDG-uptake in liver [32,33], and it is recommended that patients are kept euglycemic, especially when the liver is the organ of interest [34].

Moreover, in patients with fatty liver disease, which is an increasing problem worldwide, the hepatic fat content can affect the FDG-uptake in liver, as the hydrophilic FDG does not enter the fat droplets in the hepatocytes, resulting in a dilution of the signal [35]. In addition, the FDG-uptake in liver in patients with overt hypothyroidism is increased compared to euthyroid individuals [36], whereas the opposite pattern is seen in patients with hyperthyroidism [37]. Lastly, FDG-uptake in liver is also affected both by sex [38] and age [39].

As the Deauville score is a visual score, one could argue that what the eye sees, when looking at the liver is the average value of signal, that is, the liver SUVmean, and not the liver SUVmax. This could in theory mean that a visual evaluation would not be as affected by the change of acquisition times as would a quantitative assessment; however, still we find three patients with change in their (visual) DS due to increased scan duration.

The new LAFOV PET/CT systems convey many improvements with possibilities of optimization of image quality or reduction in injected radioactivity or in acquisition time. This is beneficial to many patients, including, among others, patients with malignant lymphomas as they may be young and will undergo several PET/CT scans during their life. Moreover, in children and pregnant women, the possibilities of reducing PET acquisition time or injected activity–dose are remarkable [40,41,42], and in children, the reduced scan duration even allows for scanning without sedation [43]. Furthermore, an unsolved clinical case was clarified within a 1 min scan on a LAFOV PET/CT-scanner, displaying giant cell arteritis and polymyalgia rheumatic [44], proving the capability of reducing acquisition time considerable. Others report that reducing PET acquisition time to 6 min in a LAFOV PET/CT scanner in patients with malignant melanoma was associated with absolutely no clinical potential consequences in the context of staging or restaging [45]. Even ultrafast PET/CT with acquisition times reduced right down to 30 s have been proven feasible and is especially important for patients with claustrophobia or an inability to lie down [46].

In research, LAFOV PET/CT gives many new opportunities, among others with long-life radionuclides such as Zirconium-89 (^89^Zr) used in immuno-PET, e.g., with ^89^Zr-trastuzumab in patients with HER2-positive breast cancer [47]. The increased sensitivity of LAFOV PET/CT and thus a better signal-to-noise ratio enables a substantial reduction in the amount of administered dose rendering the method more operable with regard to radiation exposure [48]. In addition, for short-lived radionuclides, e.g., ^15^O-H_2_O with a half-life of ~2 min, the LAFOV PET/CT gives opportunities for studying tracer uptake in all organs of interests, before the tracer decays, due to the long-axial FOV, thus avoiding repeated injection of the tracer.

In addition, as one bed position can cover all organs of interest from vertex to mid-thigh, the LAFOV PET/CT gives the possibility of studying whole body dynamic PET without the need for arterial cannulation, in addition to studying connections between different organ systems such as the gut-brain axis [48]. Furthermore, in the future, screening of healthy individuals with an increased risk of cancer might even be feasible.

## 5. Limitations

Our study is a single-center retrospective analysis. Involving more patients and eventually other centers would have given a more robust result. In addition, a large part of our patients was assigned DS 1 with no measurable residual lymphomas, giving us fewer data to analyze. Thus, a validation of our results in a larger trial is recommended.

## 6. Conclusions

The new LAFOV PET/CT systems present many new opportunities both in research and in clinical practice. However, the improvements in signal-to-noise ratio also convey changes that could have clinical implications not to be neglected, including impact on visual scoring systems.

Attention should be drawn towards the potential reduction of noise in the liver parenchyma when increasing acquisition times, which can translate into a higher DS and thereby have clinical implications. In our study, the influence on DS was smaller and without clinical implications when using OSEM reconstruction.

We believe that further studies are needed to decide which reconstruction method and acquisition time is optimal for assessing treatment response in patients with FDG-avid lymphomas. In our institution, we prefer using OSEM reconstruction of 300 s images for assessing DS to reduce the impact of noise on the DS. In the future, the SUVmean of the liver might be the preferred reference standard for assignment of DS.

## Figures and Tables

**Figure 1 diagnostics-13-00947-f001:**
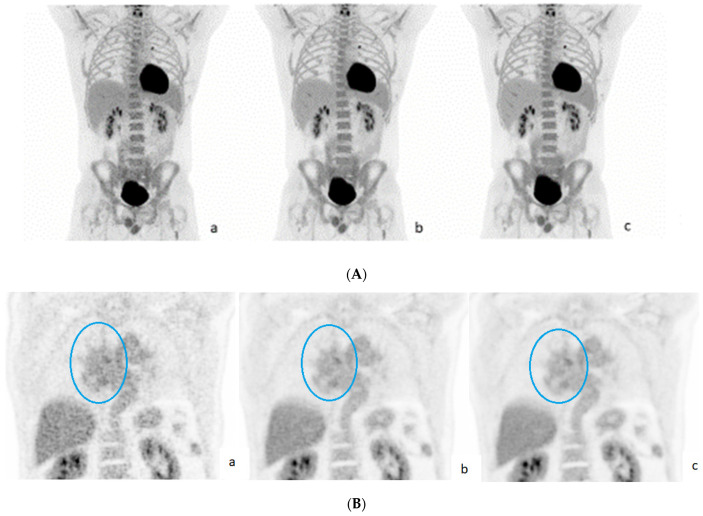
(**A**). OSEM+PSF reconstruction for acquisition times of 90 s (**a**), 300 s (**b**), and 600 s (**c**). Noise in liver is visually clearly reduced with longer scan times. (**B**). OSEM+PSF reconstruction for acquisition times of 90 s (**a**), 300 s (**b**), and 600 s (**c**). Patient with DLCBL; residual lymphoma in mediastinum is marked with a blue circle. Like in Figure 1A, noise in liver is visually clearly reduced with longer scan times with potential impact on visual scoring.

**Figure 2 diagnostics-13-00947-f002:**
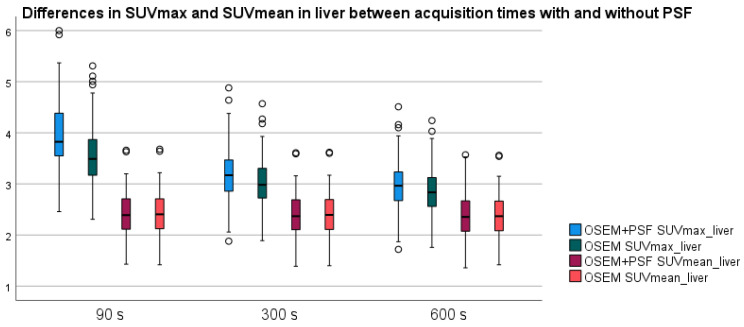
SUVmax and SUVmean in liver; differences between acquisition times with and without PSF. SUVmax in liver decreases with increasing acquisition times, whereas SUVmean is stable over time.

**Figure 3 diagnostics-13-00947-f003:**
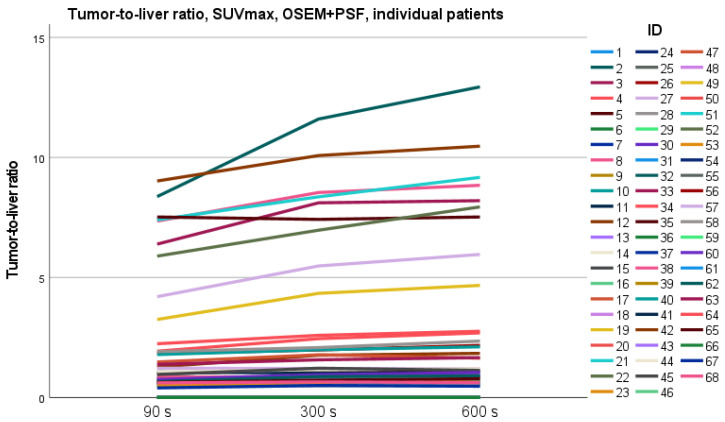
Tumor SUVmax/liver SUVmax ratio increases with increasing acquisition times due to decreasing SUVmax in liver.

**Figure 4 diagnostics-13-00947-f004:**
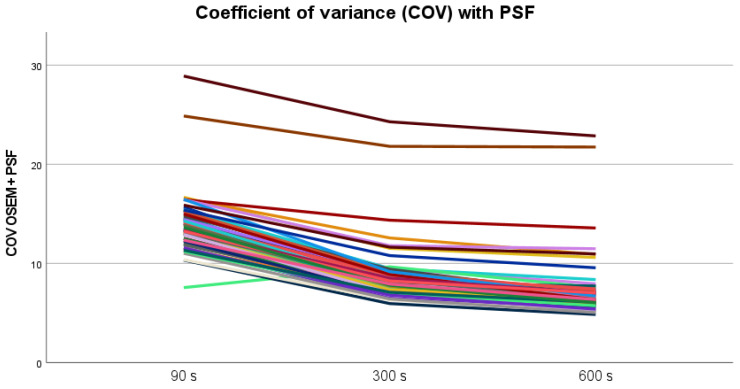
COV in liver parenchyma reduces with increasing acquisition time both for series with and without PSF. Two outliers were seen with both reconstruction methods.

**Table 1 diagnostics-13-00947-t001:** Clinical characteristics of the patients.

Total number of patients	68
Male	38
Female	30
Mean age ± SD	63.4 years ± 16.9
Age range	23–86 years
Diagnosis	
Hodgkin’s lymphoma	9
DLBCL	35
Follicular lymphoma	8
T-cell lymphoma	4
B-cell lymphoma	9
Other non-Hodgkin’s lymphoma	3

**Table 2 diagnostics-13-00947-t002:** Visual analysis of the Deauville score (DS).

	OSEM+PSF 90 s	OSEM+PSF 300 s	OSEM+PSF 600 s	OSEM 90 s	OSEM 300 s	OSEM 600 s
DS 1 (N)	23	23	23	23	23	23
DS 2 (N)	17	17	17	20	20	20
DS 3 (N)	7	5	5	3	3	3
DS 4 (N)	8	9	9	9	8	8
DS 5 (N)	13	14	14	13	14	14
Total	68	68	68	68	68	68

## Data Availability

Data cannot be made publicly available for ethical and legal reasons, as public availability would compromise patient confidentiality, and local (Danish) legislation prohibits public availability of the data.

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
