# Peer review of "Impact of Reduced Image Noise on Deauville Scores in Patients with Lymphoma Scanned on a Long-Axial Field-of-View PET/CT-Scanner"

_diagnostics, 2023, doi:10.3390/diagnostics13050947_

Round 1

Reviewer 1 Report

Dear authors,

this is a really interesting work focusing on an open issue of nuclear medicine as you underline in the introduction and in the discussion.

The english language and the syntax are good.

The introduction clearly underlines the objective of the study and the position of the work in the current scientific scenario. The section regarding material and methods is accurate and provides all relevant informations. Results and discussion sections are clearly exposed.

Images and tables are well designed and help understanding the meaning of the work.

Author Response

Dear Reviewer 1

Thank you for your kind comments on our work.

We are very happy that you find the subject of our work interesting. 

Reviewer 2 Report

The article is interesting and well conducted; it adds to other opinions on the Deawill score, thus increasing the experiences on this topic.

The only note worthy of mentioning is that I have verified that the bibliographic entries are rather dated: almost 60% refer to years prior to 2018. It would be good to find more recent entries given the topicality of the method. In particular item no. 24 should be cited as follows: "Zwezerijnen, G.J.C., Eertink, J.J., Ferrández, M.C. et al. Reproducibility of [18F]FDG PET/CT liver SUV as reference or normalization factor. Eur J Nucl Med Mol Imaging 50, 486–493 (2023). https://doi.org/10.1007/s00259-022-05977-5."

But other recent publications can be consulted.

Author Response

Dear Rewiewer 2

Thank you very much for reviewing our work and for your sensible suggestions and comments.

We have gone through our citations, and item no. 24 is now corrected to: "Zwezerijnen, G.J.C., Eertink, J.J., Ferrández, M.C. et al. Reproducibility of [18F]FDG PET/CT liver SUV as reference or normalization factor. Eur J Nucl Med Mol Imaging 50, 486–493 (2023). https://doi.org/10.1007/s00259-022-05977-5."

In addition the following recent publications have been added: 

Eskian, M., et al., Effect of blood glucose level on standardized uptake value (SUV) in (18)F- FDG PET-scan: a systematic review and meta-analysis of 20,807 individual SUV measurements. Eur J Nucl Med Mol Imaging, 2019. 46(1): p. 224-237.

Keramida, G. and A.M. Peters, FDG PET/CT of the non-malignant liver in an increasingly obese world population. Clin Physiol Funct Imaging, 2020. 40(5): p. 304-319.

Keramida, G. and A.M. Peters, The appropriate whole body metric for calculating standardised uptake value and the influence of sex. Nucl Med Commun, 2019. 40(1): p. 3-7.

Cao, Y., et al., Age-related changes of standardized uptake values in the blood pool and liver: a decade-long retrospective study of the outcomes of 2,526 subjects. Quant Imaging Med Surg, 2021. 11(1): p. 95-106.

Reviewer 3 Report

  The authors evaluated the impact of reduced image noise on Deauville scores in patients with lymphoma scanned on a long-axial field-of-view PET/CT-scanner. The study seems novel and scientifically sound. The results are interesting and worthy of publication. Some questions arise: - can this technique be used to assess prognosis in patients suffering from lymphoma? Can this method emerge as a new prognostication index in lymphoma? - the authors need to clarify how metabolic disorders (obesity, diabetes etc) impact the findings   
